# Acquired AmpC β-Lactamases among *Enterobacteriaceae* from Healthy Humans and Animals, Food, Aquatic and Trout Aquaculture Environments in Portugal

**DOI:** 10.3390/pathogens9040273

**Published:** 2020-04-09

**Authors:** Teresa Gonçalves Ribeiro, Ângela Novais, Elisabete Machado, Luísa Peixe

**Affiliations:** 1UCIBIO-REQUIMTE, Laboratory of Microbiology, Faculty of Pharmacy, University of Porto, 4050-313 Porto, Portugal; teresampg84@gmail.com (T.G.R.); aamorim@ff.up.pt (Â.N.); elisabetemac@gmail.com (E.M.); 2FP-ENAS/CEBIMED, Faculty of Health Sciences, University Fernando Pessoa, 4200-250 Porto, Portugal

**Keywords:** CMY-2, *Escherichia coli*, ST48, ST665, plasmids

## Abstract

We aimed to investigate the occurrence of acquired AmpC β-lactamases (qAmpC), and characterize qAmpC-producing *Enterobacteriaceae* from different non-clinical environments in Portugal. We analysed 880 *Enterobacteriaceae* resistant to third-generation cephalosporins recovered from 632 non-clinical samples [healthy human and healthy animal (swine, chickens) faeces; uncooked chicken carcasses; aquatic and trout aquaculture samples]. Bacterial and qAmpC identification, antibiotic susceptibility, clonal (PFGE, MLST) and plasmid (S1-/I-*Ceu*I-PFGE, replicon typing, hybridization) analysis were performed using standard methods. The occurrence of qAmpC among *Enterobacteriaceae* from non-clinical origins was low (0.6%; n = 4/628 samples), corresponding to CMY-2-producing *Escherichia coli* from three healthy humans (HH) and one uncooked chicken carcass (UCC). We highlight a slight increase in CMY-2 human faecal carriage in the two periods sampled [1.0% in 2013–2014 versus 0% in 2001–2004], which is in accordance with the trend observed in other European countries. CMY-2-producing *E. coli* belonged to B2_2_-ST4953 (n = 2, HH), A_0_-ST665 (n = 1, HH) or A_1_-ST48 (n = 1, UCC) clones. *bla*_CMY-2_ was identified in non-typeable and IncA/C_2_ plasmids. This study is one of the few providing an integrated evaluation of the qAmpC-producing *Enterobacteriaceae* occurrence, which was low, from a very large collection of different non-clinical origins. Further surveillance in contemporary collections can provide an integrated epidemiological information of potential shifts in reservoirs, transmission routes and mechanisms of dissemination of *bla*_qAmpC_ in non-clinical settings.

*Enterobacteriaceae* resistant to third-generation cephalosporins are endemic in many parts of the world, mainly by the production of extended-spectrum β-lactamases (ESBLs) or acquired AmpC β-lactamases (qAmpC) [1]. While the occurrence and characterization of ESBL-producers across different niches has been characterized in detail, far less data concerning qAmpC-producing *Enterobacteriaceae* are available. In fact, existing studies have been focused particularly on clinical niches, and/or specific bacterial species (*Escherichia coli*, non-typhoidal *Salmonella*) and/or qAmpC-types (mostly CMY-2) [2,3], while the occurrence of qAmpC-producing *Enterobacteriaceae* across different non-clinical settings is barely known.

In Portugal, DHA-1 and CMY-2 are the most commonly found qAmpC-types among *Enterobacteriaceae* associated with hospital- or community-acquired human infections, although a recent increase in CMY-2 was observed in recent years (44%, 2010–2013 versus 6%, 2002–2008) [1]. Since *bla*_CMY-2_ is the most common qAmpC-type among non-clinical niches [3], we aimed to investigate the occurrence of qAmpC producers among a very large collection of *E**nterobacteriaceae* isolates resistant to third-generation cephalosporins, recovered from a wide range of non-clinical samples, spanning the periods described above [1].

We analysed 880 *Enterobacteriaceae* isolates resistant to third-generation cephalosporins recovered from different non-clinical samples (n = 628): (i) healthy human faeces; (ii) healthy livestock animal faeces (78 swine, 44 chickens), (iii) uncooked chicken carcasses for human consumption, (iv) water/wastewater samples of aquatic environments, (v) and water/sediment/feed samples of trout aquaculture environments (Figure 1). Samples were collected from different regions of Portugal, and processed as described [4,5,6,7]. Briefly, the aliquots of uncooked chicken carcasses (0.2 mL) and aquaculture samples (0.1 mL) pre-enriched in buffered peptone water, and aliquots of the remaining samples (0.2 mL of human or animal faeces, 0.1 mL of wastewaters, filters from filtered water of aquatic environments) were seeded on CHROMagarTM Orientation/MacConkey agar plates supplemented with ceftazidime (1 mg/L) or cefotaxime (1 mg/L) [4,5,6,7]. Representative *Enterobacteriaceae* isolates (oxidase negative, approximately one to five unique morphotypes per plate) were recovered from the agar plates supplemented with antibiotics. qAmpC producers were preliminary identified by phenotypic criteria and/or polymerase chain reaction and the sequencing of genes coding for qAmpC enzymes (CMY, MOX, FOX, LAT, ACT, MIR, DHA, ACC) [1]. Bacterial identification, antibiotic susceptibility testing, and *bla*_q__AmpC_ genetic context were performed in qAmpC positive isolates as previously described [1]. Clonal relatedness was investigated by *Xba*I-PFGE and MLST (http://enterobase.warwick.ac.uk/species/index/ecoli), and *E. coli* phylogenetic groups were identified [1]. Plasmid analysis included replicon typing, S1-/I-*Ceu*I-PFGE and hybridization [1].

We identified four CMY-2-producing *E. coli* recovered from three healthy humans [1 female (aged 65) and 2 males (aged 25 and 67)], and one uncooked chicken carcass (Table 1). This represents a very low occurrence (0.6%; n = 4/628 samples) of qAmpC producers in the samples analysed. It is of note that, although the human faecal carriage rate of qAmpC-producers was low, a slight increase between 2001–2004 (0%) and 2013–2014 (1.0%; n = 3/312) was identified, possibly suggesting a higher colonization density that should be monitored. The current rate (1.0%) is comparable to those reported in a few other European countries [8].

The absence of qAmpC producers among livestock animals (swine, chickens) is surprising since livestock animals are known as reservoirs of *bla*_CMY-2_ [3], which might be explained by the low number of analysed samples from each period. However, the detection of CMY-2-producing *E. coli* in an uncooked chicken carcass (n = 1/20; 5.0%), is of note since it can have either animal origin or cross-contamination of meat by humans during processing or at retail. In any case, this rate can be underestimated considering the low number of samples tested, and highlights the need of further monitorization in animals or at retail settings. To the best of our knowledge, this study represents the first analysing such a diverse and representative sample from aquatic environments, with the absence of qAmpC producers suggesting a particularly low burden in the Portuguese setting. 

CMY-2-producing *E. coli* belonged to phylogroups B2_2_ (n = 2, HH), A_0_ (n = 1, HH) or A_1_ (n = 1, UCC) (Table 1). The indistinguishable PFGE pattern found among B2-ST4953 *E. coli* isolates recovered from two-family related individuals (husband and wife), indicates a common source for CMY-2 faecal carriage or human-to-human transmission. The other two A-*E. coli* isolates were assigned to ST665 (A_0_) and ST48 (A_1_), the latter belonging to the clonal complex 10. These clones have been sporadically identified as CMY-2 producers from clinical (ST665 in South Africa) or non-clinical (ST48 in Tunisia and Poland) settings [9,10,11], or associated with other mechanisms of antibiotic resistance (e.g., ESBL/carbapenemase production or plasmid-mediated colistin resistance) in different reservoirs (hospitals, animals or food products) [11,12], further supporting circulation of these clones along the food chain. All the CMY-2-producing *E. coli* isolates were multidrug-resistant (non-susceptible to at least one agent in three or more antimicrobial categories tested). CMY-2 was identified in frequently reported genetic contexts (ΔIS*Ecp1*/IS*Ecp1*-*bla*_CMY-2_-*blc*-*sugE*) (1) (Table 1). When characterizable, plasmids carrying *bla*_CMY-2_ were identified as IncA/C_2_ (150 kb) (Table 1), a plasmid type previously associated with *bla*_CMY-2_ among clinical isolates in our country [1]. 

To the best of our knowledge, this is one of the few studies that determined the occurrence of qAmpC-producing *Enterobacteriaceae* among environmental, animal and human niches. Our data confirm the presence of qAmpC producers outside the clinical setting in our country and highlight the need for further surveillance in contemporary collections, that will provide integrated epidemiological information on potential shifts in reservoirs, trends of qAmpC-producers and *bla*_qAmpC_, and transmission routes among different reservoirs.

## Figures and Tables

**Figure 1 pathogens-09-00273-f001:**
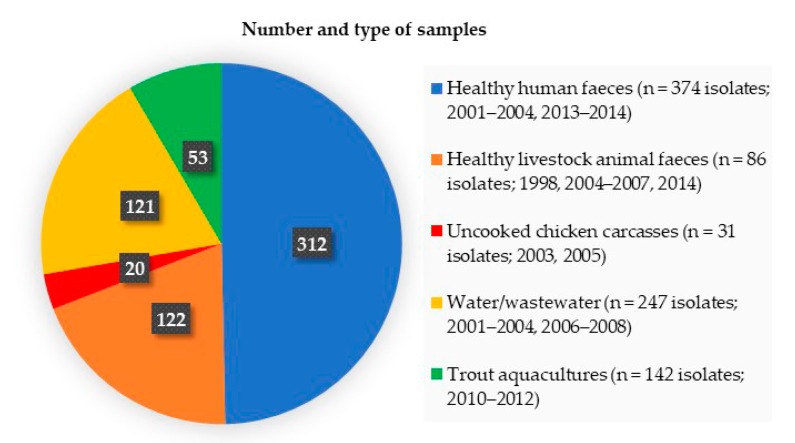
Number and type of samples (number *of Enterobacteriaceae* isolates resistant to third-generation cephalosporins; year of collection) analysed in this study.

**Table 1 pathogens-09-00273-t001:** Features of CMY-2-producing *E. coli* recovered from non-clinical origins in Portugal.

Origin (Sample) ^a^	Year of Collection	PhG ^b^	ST/CC/PFGE-Type ^c^	Genetic Environment of *bla*_CMY-2_	Plasmid Replicon Content [Inc Family (Size in kb)]		Resistance to Non-β-Lactams ^f^
					Associated with *bla*_CMY-2_	Other	
HH (33)	2014	B2_2_	ST4953/EC2	ΔIS*Ecp1*-*bla*_CMY-2_-*blc*-*sugE*	ND ^d^	FII + I1	GEN, NET, TOB, STR, TET, CHL, NAL, CIP, SUL
HH (34)	2014	B2_2_	ST4953/EC2	ΔIS*Ecp1*-*bla*_CMY-2_-*blc*-*sugE*	ND ^d^	FII	GEN, NET, TOB, STR, TET, CHL, NAL, CIP, SUL
HH (97)	2014	A_0_	ST665/EC3	IS*Ecp1*-*bla*_CMY-2_-*blc*-*sugE*	NT ^e^ (75)	K + B/O	STR, TET, NAL, SUL
UCC (6)	2003	A_1_	ST48/CC10/EC4	IS*Ecp1*-*bla*_CMY-2_-*blc*-*sugE*	A/C_2_ (150)	-	KAN, GEN, TOB, STR, TET, CHL, SUL

^a^ HH, healthy humans; UCC, uncooked chicken carcass; ^b^ PhG, *E. coli* phylogenetic group; ^c^ ST, Sequence Type, CC, clonal complex; ^d^ ND, not determined due to multiple plug degradations; ^e^ NT, non-typeable; ^f^ CIP, ciprofloxacin; CHL, chloramphenicol; GEN, gentamicin; KAN, kanamycin; NAL, nalidixic acid; NET, netilmicin; STR, streptomycin; SUL, sulphonamides; TET, tetracycline; TOB, tobramycin.

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
