# Peer review of "Acquired AmpC β-Lactamases among Enterobacteriaceae from Healthy Humans and Animals, Food, Aquatic and Trout Aquaculture Environments in Portugal"

_pathogens, 2020, doi:10.3390/pathogens9040273_

Round 1

Reviewer 1 Report

I think the paper is OK. My one suggestion is to ask the authors to add a bit of detail regarding sample selection -- even if it is described in detail in other papers -- so that the reader can assess the potential for selection bias.

Otherwise it's ok.

Author Response

Point 1: I think the paper is OK. My one suggestion is to ask the authors to add a bit of detail regarding sample selection -- even if it is described in detail in other papers -- so that the reader can assess the potential for selection bias. Otherwise it's ok.

Response 1: We have included details regarding sample processing that can be found in this new manuscript version (lines 56 to 60). Details about number of samples, number of isolates and year of collection, are illustrated in Figure 1.

Reviewer 2 Report

Authors responded to my comment.

Author Response

The reviewer 2 did not suggest any modifications.

This manuscript is a resubmission of an earlier submission. The following is a list of the peer review reports and author responses from that submission.

Round 1

Reviewer 1 Report

The authors describe a study aimed at measuring the occurance of 3 Gen Cepholosporin resistance in enterobacteriaceae recovered from different sources in Portugal. The goal of the study was to identify potential reservoirs of these organisms. While this is an important objective, I don't think it was achieved in this study.

My main reservation about this work is that there is no clear sampling plan. Most of the samples (56%) were from healthy humans, so it is not surprising that 3/4 isolates recovered were from this source. It was not clear how the other sites were selected, and samples sizes from the other sources were fairly small, so I don't think it is possible to make any conclusions about potential sources of 3GC resistant organisms based on this work. The description of the methods is necessarily long, but still not detailed enough for readers to gain a clear understanding of the difference in methods used for recovery of isolates from different source (i.e. limit of detection? What's the rationale for pre-enrichment of some samples and not others?). I'm not sure you can do molecular epidemiology based on recovery of 4 strains.

With only 4, samples recovered, it is not clear that the objective of measuring diversity of blaAmpC-producing enterobacteriaceae has been met.

I did not spend the time to go over writing, but work needs to be done to improve language/clarity. Some specific examples:

Line 32 " with resistance occurring mainly by the production of..."
Line 35 "hindering the identification of transmission..."
Line 36 "have been focused on particular niches"
Line 54: how can you get 200 mL from a 1 mL suspension. How many g of feces in 1 mL suspension?

Reviewer 2 Report

Major comments

This paper is a useful contribution to the AMR literature.

While the diversity in the study is appreciated, the origin (no. of samples), sample types, and years of collection need to be better justified.  At present the reader is left wondering, for example, why samples were taken over such periods. Was there a logic to it?

Also, is data missing from Table 2? (The number of different samples mentioned in Table 1 suggests that there should be many more findings, no?)

Minor comments

Should define multidrug resistance on line 109

Needs some language editing.

Needs some in-text citation editing.

While I am generally in favour of very flexible section division (or none at all) I would recommend adding 'Methods' before Line 50 and 'Findings' before Line 76.

Reviewer 3 Report

Dear Authors,

Your manuscript provides great insight molecular epidemiology of qAmpC-producing Enterobacteriaceae. However, I would recommend revising abstract part to directly indicate finding of the study.